# Dissemination of *Isaria fumosorosea* Spores by *Steinernema feltiae* and *Heterorhabditis bacteriophora*

**DOI:** 10.3390/jof6040359

**Published:** 2020-12-11

**Authors:** Jiří Nermuť, Jana Konopická, Rostislav Zemek, Michal Kopačka, Andrea Bohatá, Vladimír Půža

**Affiliations:** 1Department of Biodiversity and Conservation Biology, Institute of Entomology, Biology Centre CAS, Branišovská 1160/31, 370 05 České Budějovice, Czech Republic; jana.konopicka@entu.cas.cz (J.K.); michal258@centrum.cz (M.K.); vpuza@seznam.cz (V.P.); 2Faulty of Agriculture, University of South Bohemia in České Budějovice, Studentská 1668, 370 05 České Budějovice, Czech Republic; abohata@centrum.cz; 3Arthropod Ecology and Biological Control Research Group, Ton Duc Thang University, Ho Chi Minh City 758307, Vietnam; rostislav.zemek@tdtu.edu.vn; 4Faculty of Applied Sciences, Ton Duc Thang University, Ho Chi Minh City 758307, Vietnam; 5Department of Biochemistry and Physiology, Institute of Entomology, Biology Centre CAS, Branišovská 1160/31, 370 05 České Budějovice, Czech Republic

**Keywords:** entomopathogenic nematodes, entomopathogenic fungi, conidia, blastospores, dispersal, PDA plates, soil substrate

## Abstract

Entomopathogenic nematodes and fungi are globally distributed soil organisms that are frequently used as bioagents in biological control and integrated pest management. Many studies have demonstrated that the combination of biocontrol agents can increase their efficacy against target hosts. In our study, we focused on another potential benefit of the synergy of two species of nematodes, *Steinernema feltiae* and *Heterorhabditis bacteriophora,* and the fungus *Isaria fumosorosea*. According to our hypothesis, these nematodes may be able to disseminate this fungus into the environment. To test this hypothesis, we studied fungal dispersal by the nematodes in different arenas, including potato dextrose agar (PDA) plates, sand heaps, sand barriers, and glass tubes filled with soil. The results of our study showed, for the first time, that the spreading of both conidia and blastospores of *I. fumosorosea* is significantly enhanced by the presence of entomopathogenic nematodes, but the efficacy of dissemination is negatively influenced by the heterogeneity of the testing arena. We also found that *H. bacteriophora* spread fungi more effectively than *S. feltiae*. This phenomenon could be explained by the differences in the presence and persistence of second-stage cuticles or by different foraging behavior. Finally, we observed that blastospores are disseminated more effectively than conidia, which might be due to the different adherence of these spores (conidia are hydrophobic, while blastospores are hydrophilic). The obtained results showed that entomopathogenic nematodes (EPNs) can enhance the efficiency of fungal dispersal.

## 1. Introduction

Entomopathogenic nematodes (EPNs) of the families Steinernematidae Filipjev, 1934, and Heterorhabditidae Poinar, 1976, are ubiquitous soil organisms and can be found in almost all terrestrial ecosystems except for Antarctica. These nematodes live in obligate association with bacteria of the genera *Xenorhabdus* Thomas and Poinar, 1979, and *Photorhabdus* Boemare et al., 1993, (Enterobacterales: Morganellaceae) [1] and infect and kill a wide range of insects [2]. The only free-living stage is infective juveniles (IJs), which can detect hosts over long distances due to their sense organs. They are able to react to many host-associated cues, such as carbon dioxide, arginine, uric acid, and ammonia [3,4,5,6,7]. When a signal from a host is detected, an IJ starts to move against the gradient of this chemical. When they reach the host, they invade its body, release bacteria into the hemolymph, causing septicemia, and the host usually dies within 24 or 48 h. The ability to actively search for hosts and rapidly kill them make EPNs good biocontrol agents.

Some EPN species are mass cultured in liquid media or in surrogate hosts (e.g., *Galleria mellonella* Linnaeus, 1758 (Lepidoptera: Pyralidae) larvae) and widely used in biological control to reduce the damage caused by soil-dwelling pestiferous insects, e.g., grubs, caterpillars, and dipteran larvae [8]. Usually, these nematodes have a wide range of hosts and can be used to control populations of many insects; however, some nematodes (e.g., *Steinernema scapterisci* Nguyen and Smart, 1990 (Rhabditida: Steinernematidae) show an affinity for and a greater ability to manage populations of one pest (mole crickets) or a group of pests (e.g., grubs are controlled by *Heterorhabditis bacteriophora* Poinar, 1976 (Rhabditida: Heterorhabditidae)) [8].

Entomopathogenic fungi (EPFs) are common in nature, have a cosmopolitan distribution and cause natural epizootics in populations of insects or other arthropods [9,10,11]. EPFs belonging to the order Hypocreales have a very wide range of hosts. The most important species include *Beauveria bassiana* (Bals.-Criv.) Vuill, 1912, *Isaria fumosorosea* Wize, 1904 and *Metarhizium anisopliae* (Metchnikoff) Sorokin, 1883 (Hypocreales: Cordycipitaceae) [12]. EPFs are unique organisms that are capable of infecting their hosts directly through the exoskeleton, while other entomopathogens (viruses and bacteria) must be ingested with food to infect the host [13]. The process of host infection starts with the adhesion of the fungal conidia to the host cuticle and then involves conidial germination, the differentiation of germ tubes into appressoria, cuticle penetration, hyphal differentiation into blastospores in the hemolymph inside the insect body, host colonization, and hyphal proliferation onto the host cadaver surface [12,14,15]. Under suitable conditions, the fungus grows on the surface of the body of the dead host and sporulates. A secondary generation of conidia is then released into the environment [16]. Conidia can be spread in the environment in several ways, e.g., by weather factors (wind and rain) or by biotic vectors that can spread spores over long distances [17,18]. Common mechanisms of the horizontal transmission of fungal diseases in invertebrate populations include the contact of healthy individuals with infected cadavers and the autodissemination of conidia within populations in connection with specific intrapopulation processes, such as migration, copulation, and oviposition [12,19]. Entomopathogenic fungi persist in the soil for a long time [20,21,22], but the process of spore dispersion is limited [10]. Entomopathogenic fungi from the Hypocreales order are already promising candidates in practical biological control against serious pests [23,24]. Numerous commercial products based on EPFs are registered around the world. The dominant species incorporated into these bioproducts are *B. bassiana* and *M. anisopliae*, which represent almost 70% of all bioproducts [24]. The fungi are produced either in solid-state fermentation, where they produce aerial conidia, or in a submerged liquid-state fermentation, where they produce blastopores [25]. The EPF *I. fumosorosea*, which is widely spread and has a wide host range, is one of the most prominent species of EPF [26]. This fungus can cause natural insect epizootics in populations of sucking pests [23,27]. Several commercial bioproducts based on various strains of *I. fumosorosea* have been successfully used in biocontrol, especially in Europe and the USA [26,28].

The efficacy of biocontrol agents can be improved by their combination [29,30,31]. Entomopathogenic fungi and entomopathogenic nematodes applied together performed more efficiently than when applied alone [32,33,34,35]. In contrast, Shapiro-Ilan et al. [36] found that when pairs of nematode and fungal pathogens attacked the larvae of the weevil *Curculio caryae* Horn, 1873 (Coleoptera: Curculionidae), most pairings were less effective than a single highly effective entomopathogenic species. Although nematodes do not seem to be directly affected by EPFs, Hussein et al. [35] found that when *Steinernema feltiae* Filipjev, 1934 (Rhabditida: Steinernematidae) was applied to Colorado potato larvae 24 h or later after *I. fumosorosea*, the penetration rate and the development of the nematodes were negatively affected. A similar negative effect was observed in the interactions between the fungus *M. anisopliae* and nematodes *Steinernema glaseri* Steiner, 1929 (Rhabditida: Steinernematidae) [37] and *H. bacteriophora* [38] and between *B. bassiana* and *Steinernema ichnusae* Tarasco et al., 2008 (Rhabditida: Steinernematidae) [39].

Invertebrates can effectively vectorize fungal spores and increase the chances of contact between them and their hosts. In particular, there is high potential in pollinators [40,41,42] or predators that have been recently investigated in several studies. Zhang et al. [43] reported that predatory mites can spread conidia of *B. bassiana*. A method has been developed in which the soil predatory mite *Stratiolaelaps scimitus* Berlese (Acari: Laelapidae) and the plant-inhabiting mites *Neoseiulus cucumeris* Oudemans and *Amblyseius swirskii* Athias-Henriot (Acari: Phytoseiidae) collect and transport *B. bassiana* conidia directly from commercial rearing substrates to control western flower thrips, *Frankliniella occidentalis* Pergande, 1895 (Thysanoptera: Thripidae) [44]. To our knowledge, the dissemination of spores of entomopathogenic fungi by EPNs has not yet been studied.

The main goal of this study was to evaluate the possibility of the dispersal of spores (both conidia and blastospores) of the EPF *I. fumosorosea* by the EPNs *S. feltiae* and *H. bacteriophora*. We not only targeted the differences between the different types of spores and different vector species but also tried to estimate the effects of environmental heterogeneity (experimental arenas) and the desheathment of the IJs.

## 2. Materials and Methods

### 2.1. Experimental Organisms

The entomopathogenic nematodes *Steinernema feltiae* strain NFUST (isolated during the 1980s near the town of Izhevsk, Russia) and *H. bacteriophora* strain HB221 (isolated in 1997, near the village of Pouzdřany, Czech Republic) were used for this study. Both strains are maintained in the nematode collection of the Laboratory of Entomopathogenic Nematodes, Institute of Entomology, Biology Centre CAS in České Budějovice, Czech Republic.

The nematodes were cultured in vivo in *Galleria mellonella* (Lepidoptera: Pyralidae) larvae, which were infected with ca. 50 IJs of the appropriate nematode per larva. The dead larvae were placed on White traps [45] and newly released IJs were collected from the water. The IJs were stored for later use in sterile tap water on 90 mm Petri dishes in a refrigerator at 5 °C in the dark.

For this study, strain CCM 8367 of *Isaria fumosorosea* (Hypocreales: Cordycipitaceae) was used. The strain originated from infected pupae of the horse chestnut leaf miner, *Cameraria ohridella* Deschka and Dimić, 1986 (Lepidoptera: Gracillariidae), collected in the Czech Republic [46]. The strain was deposited as a patent culture in the Czech Collection of Microorganisms (CCM) in Brno [47,48].

*Isaria fumosorosea* was grown on Petri dishes (ø 90 mm) containing potato dextrose agar (PDA, Sigma-Aldrich, Munich, Germany, 39 g/L). The plates were incubated at 25 °C in the dark for 10–14 days. The aerial conidia were harvested by scraping them into a sterile solution of 0.05% (*v*/*v*) Tween 80^®^ (Sigma-Aldrich, Germany). The conidial suspension was filtered through sterile gauze to separate the mycelium and clusters of conidia. In the uniform suspension, the number of conidia was counted with a Neubauer improved counting chamber (Sigma-Aldrich, Germany), and subsequently, the suspension was adjusted to a concentration of 1.5 × 10^8^ spores per mL. The blastospores of the fungus were produced in potato dextrose broth (PDB, Sigma-Aldrich, Germany). For submerged cultivation, 95 mL of sterile liquid PDB medium in a 250 mL Erlenmeyer flask was inoculated with 5 mL of conidial suspension at a concentration of 1.0 × 10^7^ spores per mL. The flask was placed in an orbital shaker (Kühner AG, Birsfelden, Switzerland), and the fungus was incubated at a speed of 200 rpm at 25 °C. The blastospores were harvested after 4 days of incubation. The suspension was filtered through sterile gauze to remove the mycelia. The blastospore concentration was determined by a Neubauer improved counting chamber and adjusted to 1.5 × 10^8^ spores per mL.

### 2.2. Agar Plate Experiments

The ability of EPNs to spread the spores of an EPF was observed on PDA in 90 mm Petri dishes. To simulate the effect of heterogeneity of space in contrast with clean agar we used three different types of experimental arenas: (1) a clean 3.9% PDA plate, (2) a Petri dish with a line of silver sand acting as a barrier across the center and (3) a Petri dish with PDA and a small heap of 0.8 mm sterile silver sand in the center (Figure 1). Sand grains were considered as obstacles which are likely to affect movement of nematodes and can remove some fungus spores from nematodes cuticle. A pipette was used to place the nematodes and fungal spores in the dishes on the agar in the center of the Petri dish (Figure 1A), directly on the top of the silver sand heap (Figure 1H) or on the agar in the middle of half of the Petri dish (Figure 1B). For the experiment, 50 IJs of either nematode species (*S. feltiae* or *H. bacteriophora*) and 20 µL of spore suspension (either conidia or blastospores) at a concentration of 1.5 × 10^8^ per mL were used. Control dishes received fungus only. When the experiment was established, all the dishes were stored at 16 °C in the dark for 5 days. Then, photos of the dishes were taken, and the digital images were analyzed (see the next paragraph). For each combination, we used 10 experimental and 10 control dishes, and the whole experiment was repeated twice. In total, 480 dishes were used (2 nematode species × 2 types of spores × 20 dishes × 3 types of experimental arenas × 2 repetitions). Additional five dishes for each experimental design were used for evaluation of nematodes dispersal on the experimental plates and number of nematodes in different zones of the dishes (Figure 1) were counted.

To estimate the area covered with *I. fumosorosea* colonies, the individual Petri dishes were photographed by an Olympus SP-510 camera. The digital images were manually cleaned, and the colors were inverted to obtain negative images using Adobe Photoshop CS3 and Space Navigator (3Dconnexion). The resulting files were then processed in custom-made image analysis software written in the Java™ programming language [49]. This processing included thresholding to obtain a black and white image, counting the number of black pixels (these indicate fungi) and calculating the percentage of Petri dish area covered by *I. fumosorosea*. The results were stored in a CSV (comma-separated value) file along with a thresholded image to control whether the thresholds were correctly implemented.

### 2.3. Glass Tube Experiments

This experiment was performed in a simple apparatus that was constructed from one 10 cm long glass tube (with an inner diameter of 5 mm) and two 2 mL Eppendorf tubes attached to the ends of the glass tube (Figure 2). The glass tubes were lined with a piece of parchment paper (10 × 2 cm). Later, the tubes with paper were filled in with 3 mL of slightly wet sterilized brown soil. Openings (diameter of 8 mm) were made in the lids of the Eppendorf tubes, into which the glass tubes were inserted. Pieces of a synthetic cloth with calibrated apertures of 25 × 30 µm were glued onto the bottoms of the Eppendorf tube lids to protect the tubes against pollution by the soil in the glass tube. The Eppendorf tube on the application side of the glass tube was left empty, while one *G. mellonella* larva was added to the opposite Eppendorf tube (*Galleria* side) to stimulate the nematodes to move in that direction. The suspension with nematodes (500 IJs in 40 µL of water) and 20 µL of the suspension with either conidia or blastospores were pipetted directly onto the soil in the glass tubes on the application side (without *G. mellonella*). These glass tubes were stored horizontally at 16 °C in the dark for 3 days. After this time, the soil on top of the parchment paper was removed from the glass tubes and divided into three equal parts, which were used for fungal isolation and consequent evaluation. Each part contained 1 mL of the soil substrate. As a control, we used glass tubes with the fungus but without the nematodes. The experiment was performed with both *S. feltiae* and *H. bacteriophora* nematodes and with both conidia and blastospores of the fungus *I. fumosorosea*. For each combination, we used 10 experimental and 5 control tubes, and the whole experiment was repeated twice. In total, 120 glass tubes were used (2 nematode species × 2 types of spores × 15 glass tubes × 2 repetitions).

To explain the effect of nematode species on fungus dispersal and the potential role of the second-stage cuticle, the same experiment was performed with desheathed IJs of both nematode species. Infective juveniles were desheathed in 1% NaOCl for 5 min [50]. For this experiment, we used 5 experimental glass tubes for each nematode and spore combination and 10 controls. The whole experiment was repeated twice. In total, 60 glass tubes were used (2 nematode species × 2 types of spore × 5 glass tubes + 10 controls × 2 repetitions).

In order to test a possible effect of sheath on nematode movement, the same experiment without fungus was arranged both for ensheathed and desheathed nematodes. Nematodes from each part was extracted by using modified Baerman funnels [51] and counted under the stereomicroscope. For this experiment 6 glass tubes with desheathed and ensheatehed nematodes of both species were used making of 24 glass tubes in total.

After 3 days, the concentration of spores in the soil substrate was analyzed from each part obtained from the glass tubes. From all the variants, 1 mL of the soil substrate from the first part (the side where the IJs and spores were applied) was suspended in 100 mL of sterile solution of Tween 80^®^ (0.05% *v*/*v*) in a 250 mL Erlenmeyer flask. The samples were then placed on an orbital shaker for 20 min at 200 rpm at a temperature of 25 °C. After homogenization, each sample was diluted twice (1:100) with Tween 80^®^ solution. One milliliter of soil substrate from the middle part and 1 mL from the last part were suspended in 5 and 3 mL, respectively. The homogenization was performed with a Vortex (Heidolph Instruments GmbH, Schwabach, Germany). These samples were not diluted after homogenization. Each sample was transferred to a volume of 0.5 mL on the surface of a Petri dish with the selective medium dodine [52] and stirred with a sterile L-shaped spatula. The Petri dishes were incubated in a climate chamber for one week at 25 °C. After this period, the numbers of colonies of *I. fumosorosea* on the plates were counted. Three replications of each sample were prepared. The fungus concentration in the substrate was expressed as the number of colony-forming units (CFU) per milliliter of soil substrate.

### 2.4. Statistical Analyses

All the analyses were performed in the program Statistica 10 (Statsoft, Inc., Tulsa, OK, USA). Analysis of variance (ANOVA) was used for the data evaluation. Glass tube experiments were evaluated by factorial ANOVA, for agar plate experiments one-way ANOVA was used due to different (non-factorial design). All the dependent variables were logarithmically transformed to reach normality. The results of these analyses were visualized by using categorized column or box plot graphs based on raw (nontransformed) data.

## 3. Results

### 3.1. Agar Plate Experiments

The statistical analysis showed that the dispersion of *I. fumosorosea* on agar plates (Figure 3 and Figure 4) is significantly influenced by the presence of nematodes (df: 1, 48; F: 18.38; *p* < 0.001) and by the type of experimental area (df: 2, 87; F: 32.37; *p* < 0.001). The difference between dispersal of blastospores (1314 ± 700 mm^2^) and conidia (1103 ± 654 mm^2^) was not statistically significant (df: 1, 88; F: 2.27; *p* > 0.05). In general, the average fungal dispersion on the plates with *H. bacteriophora* (1219 ± 636 mm^2^) was similar (df: 1, 88; F: 0.07; *p* > 0.05) to that on the plates with *S. feltiae* (1203 ± 707 mm2). Significant differences (*p* < 0.001) were observed among the different types of experimental arenas. The highest fungal coverage (1607 ± 693 mm^2^) was recorded from the clean PDA plates with both nematodes and fungi, while in the case of the silver sand barrier (1314 ± 282 mm^2^) and heap (704 ± 654 mm^2^), the fungal dispersion was apparently lower.

According to statistical analyses there were no significant differences in the dispersal on PDA plates between *H. bacteriophora* and *S. feltiae* in any experimental arena: agar (df: 1, 54; F: 0.13; *p* > 0.05), barrier (df: 1, 72; F: 0.04; *p* > 0.05), heap (df: 1, 36; F: 1.51; *p* > 0.05). Expectedly there was recorded significant effect of the zone (Figure 1) on nematode’s dispersal: agar (df: 2, 54; F: 6.49; *p* < 0.05), barrier (df: 3, 72; F: 14.48; *p* < 0.001), heap (df: 1, 36; F: 14.86; *p* < 0.001). In general, the data showed no difference in the dispersal between *H. bacteriophora* and *S. feltiae* on the clean PDA plates and plates with sand heap or sand barrier but according to our observation the movement pattern seemed to be different as the IJs of *H. bacteriophora* were almost always cruising while the IJs of *S. feltiae* tended to remain curled up and inactive after reaching some position in the arena. This effect however was not further evaluated.

### 3.2. Glass Tube Experiment

*Isaria fumosorosea* spread within the soil substrate in the glass tubes (Table 1 as a result of its own growth; however, this dispersion was significantly enhanced by the presence of entomopathogenic nematodes (df: 1, 84; F: 5.00; *p* < 0.05). In general, *H. bacteriophora* dispersed to the distant part of the tube significantly (df: 1, 84; F: 13.82; *p* < 0.001) more spores per 1 mL of soil (64 ± 25 CFU/mL) than *S. feltiae* (11 ± 12 CFU/mL). The dispersal of the blastospores (49 ± 34 CFU/mL, 0.022 % of all recovered spores) was significantly higher (df: 1, 84; F: 7.04; *p* < 0.05) than that of the conidia (26 ± 30 CFU/mL).The presence of the second-stage cuticle on the bodies of the infective juveniles was identified as a crucial factor (df: 1, 84; F: 14.08; *p* < 0.001), because spore dissemination by desheathed IJs was almost negligible. Only in the combination treatment with *H. bacteriophora* and blastospores were 3 CFUs disseminated in one of the glass tubes (an average 0.6 CFUs per glass tube). In the other combinations, the conidia and blastospores were not disseminated. Complete results of ANOVA, interactions included, are presented in Table 2.

Statistical analyses of additional glass tube experiment for nematodes’ dispersal showed that nematodes species (df: 1, 36; F: 1.955; *p* > 0.05), presence of the second stage cuticle (df: 1, 36; F: 2.878; *p* > 0.05) and time (df: 2, 36; F: 1.144; *p* > 0.05) have no significant effect on nematodes dispersal.

## 4. Discussion

The present study explored a new alternative method for disseminating entomopathogenic fungi in the environment, mostly in the soil. The goal of this study was to evaluate and quantify the efficacy of the dissemination of conidia and blastospores of *I. fumosorosea* by *H. bacteriophora* and *S. feltiae*. We focused on the differences between the types of spores and vector species. We also estimated the effect of environmental heterogeneity and the presence of a second-stage cuticle on the IJ nematodes. Our results showed, for the first time, that EPNs can enhance the efficacy of EPF dissemination.

The utilization of other invertebrates, such as mites and insects, as vectors of fungal spores increases the chances of contact between a microorganism and its host and is considered a promising strategy for biological control programs [44]. The potential of arthropods, e.g., commercially available and mass-produced pollinators or predators, as vectors of beneficial fungi has been recently investigated in several studies [42]. Al Mazra’awi et al. [40] reported that honeybees, *Apis mellifera* Linnaeus, 1758 (Hymenoptera: Apidae), effectively dispersed *B. bassiana* conidia in canola fields, increasing the mortality of the target pest, *Lygus lineolaris* Palisot de Beauvois, 1818 (Heteroptera: Myridae), by up to five times compared to the mortality of the control. Entomovectoring systems using bumble bees are already commercially used in greenhouses [42]. Kapongo et al. [41] used *B. bassiana*, while Smagghe et al. [53] reported efficient bumble bee vectoring of *M. anisopliae* spores.

Zhang et al. [43] used two predatory mite species, *N. cucumeris* and *A. swirskii*, as vectors of *B. bassiana* conidia against *Diaphorina citri* Kuwayama (Homoptera: Psyllidae). In vitro experiments showed that the mortality rates of phytoseiid mites were between only 10% and 15%, while the mortality of *D. citri* was 100%. In assays with plants, the majority of *D. citri* individuals were killed by fungi a few days after contact with the mites, which were loaded with fungal spores. A method has been developed in which the soil predatory mite *Stratiolaelaps scimitus* and the plant-inhabiting mites *N. cucumeris* and *A. swirskii* collect and transport *B. bassiana* conidia directly from a commercial rearing substrate to control the western flower thrips, *F. occidentalis* [44]. A study by Lin et al. [54] confirmed that loading *A. swirskii* and *N. cucumeris* with *B. bassiana* can increase their capacity to suppress thrips populations by combining predation and the dispersal of pathogens. Wu et al. [55] demonstrated that another phytoseiid mite, *Neoseiulus barkeri* Hughes (Acari: Phytoseiidae), is able to disseminate *B. bassiana* conidia. Another study, however, showed that any potential benefits of fungal dissemination by predatory mites were possibly weakened by increased mite grooming time, which likely reduced the searching activity and predation rates of *N. barkeri* and suggested that the simultaneous application of *B. bassiana* and phytoseiid mites would not be recommended for effective biological control [56].

Similar to arthropods, entomopathogenic nematodes may be able to spread the spores of various fungi, and this phenomenon could explain the previously reported synergy between *I. fumosorosea* and *S. feltiae* [35]. The present study has demonstrated that although *I. fumosorosea* is able to spread in the environment through its own growth, the efficacy of its dissemination is enhanced by the presence of EPNs in the system. This effect is clearly visible both on the agar plates and in the glass tubes with soil. The fact that the fungal coverage on the agar plates and the number of CFUs in the glass tubes are higher in the presence of nematodes than in the controls suggests a certain level of synergy between the EPF and EPN, which could lead to higher efficiency against target pests, as reported by other authors [29,30,31,32,33,34]. Similarly, concomitant infection with *Steinernema diaprepesi* Nguyen and Duncan, 2002 (Rhabditida: Steinernematidae) and the saprophytic fungus *Fusarium solani* (Martius) Saccardo, 1881 (Hypocreales: Nectriaceae) increased the mortality of insects to 83% compared to 58% and 0% mortality when nematodes and fungi, respectively, were applied individually [57]. The final outcome will depend on the timing of the (co)application of the EPFs and EPNs [35], and the EPF species will probably play an important role (as in the joint application of some EPN-EPF pairs (e.g., *M. anisopliae*–*S. glaseri* and *H. bacteriophora* and *B. bassiana*–*S. ichnusae*); nevertheless, a negative effect on the efficacy has been observed [37,38,39].

In particular, from the agar plates with different treatments (clean agar, sand heap, and sand barrier), we saw that the dispersal of spores is apparently limited by spatial heterogeneity of the environment. The fungus was very effectively spread on the clean agar plates, while the sand barrier strongly limited its dissemination, probably due to the reduction in nematode movement on the plate. Only a limited number of nematodes moved across the barrier; however, the fungus was not disseminated across the barrier in almost any case. A similar effect was also observed on the plates with the sand heaps. Spore dispersal outside the heaps, on the agar surface was significantly lower than the dispersal on the clean agar. Therefore, it is possible to hypothesize that spatial heterogeneity (more abrasive environment) can limit the fungal dissemination due to removing spores from nematodes’ surface.

After observing that *H. bacteriophora* was more efficient than *S. feltiae* in transmitting spores, we formulated and tested the hypothesis that the difference could be connected to the presence of the second-stage cuticle. It is well known that *Heterorhabditis* nematodes retain their second-stage cuticle much more firmly than *Steinernema*, which usually loses this cuticle more easily [58]. Indeed, our experiment with desheathed larvae confirmed the crucial effect of the second-stage cuticle for dissemination. Generally, the presence of sheaths plays several important roles in nematode biology and survival. Sheaths can protect nematodes against desiccation [59], prevent infection with the parasitic fungus *Hirsutella rhossiliensis* Minter and Brady (Hypocreales: Ophiocordycipitaceae) [60] or reduce predation by mites [61]. The second-stage cuticle is more corrugated than the third-stage cuticle, and some parts of the sheath can be disrupted, providing the spores a good opportunity to adhere to the nematode and be transmitted. The less effective spore dispersal by the nematodes in the sand heaps and across the sand barriers could therefore be well explained by the fact that the abrasive environment could desheathed the nematodes, decreasing their ability to transmit the spores. The alternative explanation could lie in differences in foraging strategies of these two nematodes. While *H. bacteriophora* is considered typical cruiser, *S. feltiae* is intermediate species [50]. Surprisingly, no significant difference in nematodes’ dispersal was observed between these nematodes and that supports our first hypothesis about the crucial role of sheath. On the other hand, we observed, though it was not systematically evaluated, that after moving quite far from the place of application *S. feltiae* usually remained still, while *H. bacteriophora* was usually observed in active movement. Higher movement activity of *H. bacteriophora* thus could contributed to its more efficient spore dissemination. The situation in the glass tubes is very similar. There was no significant difference in number of nematodes recovered from different parts of glass tubes between *S. feltiae* and *H. bacteriophora*, but the latter dispersed more spores than *S. feltiae*. The fact that desheathed nematodes transmitted almost no spores while their movement did not differ from ensheathed nematodes in both species highlights the important role of sheath. On the other hand, *S. feltiae* was evidently able to move through the tube with soil and find the host more effectively than *H. bacteriophora* as the mortality of *Galleria* larvae and nematodes’ invasion rate in dead *Galleria* were apparently higher in *S. feltiae*. This observation highlights the role of foraging behavior in spore transmission. While *S. feltiae* goes directly and quickly to the host, *H. bacteriophora* cruise more in the soil and disperse the spores more effectively. According to our opinion and based on the results and observations both discussed factors influence the ability of the nematodes to disperse spores. In any case it will be necessary to make series of more specific experiments targeting both on the role of sheath and foraging behavior to further clarify their role in spore transmission by the nematodes.

## 5. Conclusions

This study explored a new, alternative way for dispersal of entomopathogenic fungi in the environment. The obtained results showed that EPN can enhance the efficiency of fungal dispersal. The level of fungal dissemination depends on the nematode species, spore type and heterogeneity of the environment.

## Figures and Tables

**Figure 1 jof-06-00359-f001:**
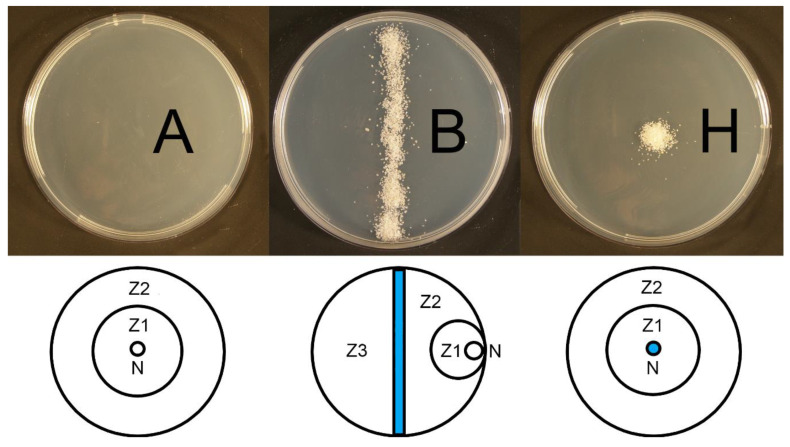
Design of the potato dextrose agar (PDA) plate experiments and zones for nematodes’ dispersal evaluation. Clean agar (**A**), silver sand barrier (**B**) and silver sand heap (**H**), Zones 1–3 (Z1–Z3), nematode or nematodes/fungus application side (N), sand marked with blue color.

**Figure 2 jof-06-00359-f002:**
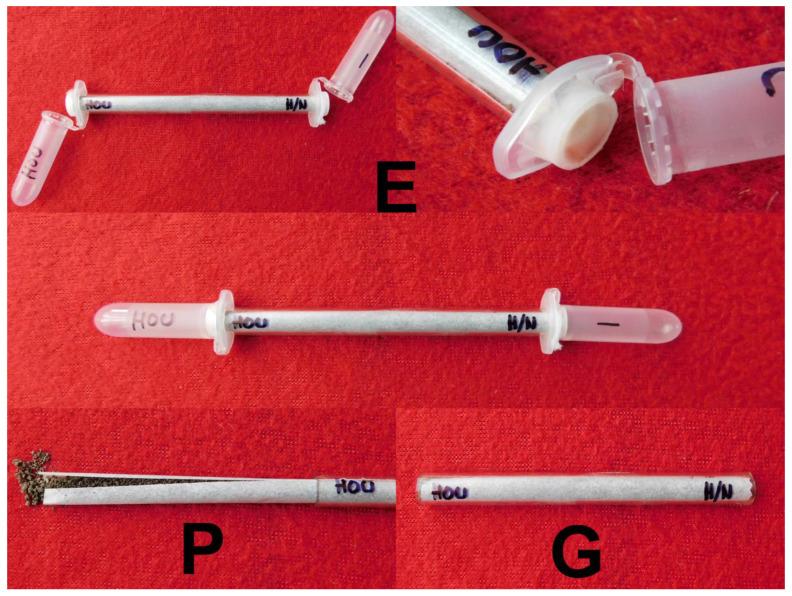
Design of the glass tube experiments. Eppendorf tubes (**E**), glass tube (**G**), parchment paper (**P**).

**Figure 3 jof-06-00359-f003:**
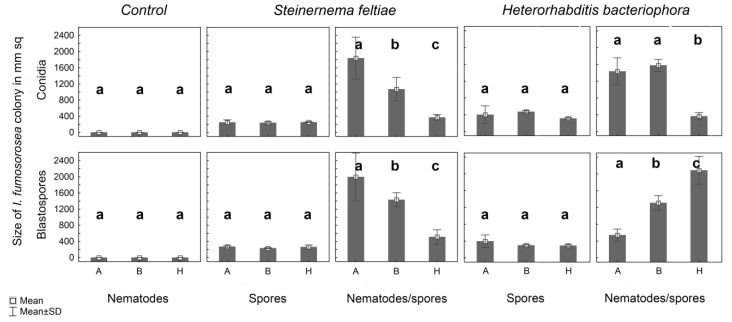
Mean (± SD) area of *Isaria fumosorosea* colonies in mm^2^. (A) Clean (pure) PDA, (B) PDA with a silver sand barrier, (H) PDA with a silver sand heap. Statistically significant differences are indicated by letters (a, b, c).

**Figure 4 jof-06-00359-f004:**
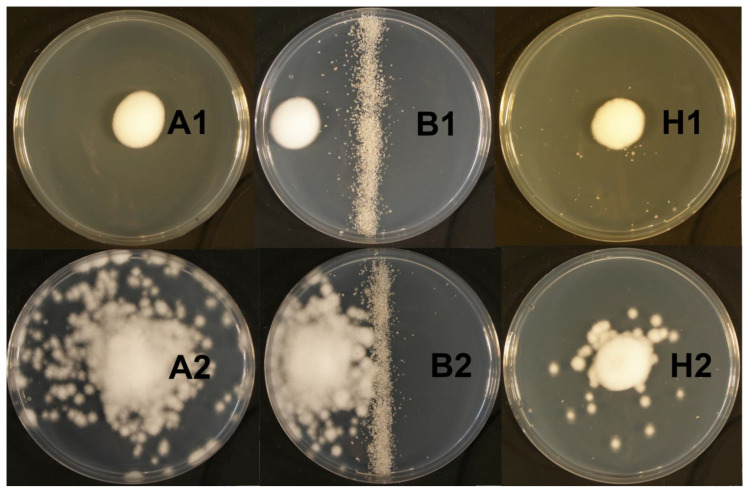
Examples of dissemination of the blastospores of *Isaria fumosorosea* by *Heterorhabditis bacteriophora* on clean PDA plates (**A**), plates with a line of silver sand as a barrier (**B**), and a sand heap (**H**). Replications were performed without nematodes (**A1**,**B1**,**H1**) and with ensheathed infective juveniles (**A2**,**B2**,**H2**). The other combinations of nematodes, spores and experimental arena type are not shown as they seem very similar. This is an illustrative photo that do not have fully correspond with data presented in Figure 3 that shows means and standard deviations.

**Table 1 jof-06-00359-t001:** Mean number of colony-forming units (CFUs) of *Isaria fumosorosea* isolated from different sections of the glass tubes with ensheathed infective juveniles of entomopathogenic nematodes. Statistically significant differences (*p* < 0.05) between “conidia or blastospores only” (control) and “conidia or blastospores + nematodes” in *Galleria* side are marked with asterisks. Dispersal of spores by desheathed nematodes is not shown, as there were almost no CFUs recovered in central and *Galleria* side.

Nematode	Spores	Glass Tube’s Side		
		Application	Centre	Galleria
*Heterorhabditis bacteriophora*	Conidia + nematodes ×	590,000	154	51
	Condia only	608,000	125	10.8
	Blastospores + nematodes ×	120,000	258	77.1
	Blastospores only	154,000	79	11.4
*Steinernema feltiae*	Conidia + nematodes	527,000	53.1	1.5
	Condia only	537,000	14.4	0
	Blastospores + nematodes ×	126,000	125	19.8
	Blastospores only	608,000	125	10.8

**Table 2 jof-06-00359-t002:** Glass tube experiments—fungi and nematodes’ dispersal: Complete results of factorial ANOVA including interactions that are not presented in the text; degrees of freedom (df), test criterion (F), *p*-value (*p*), numbers 1–4 indicate respective categorical predictors.

Spore Dispersal Experiment	Nematode Dispersal Experiment
Categorical Predictors	df	F	*p*	Categorical Predictors	df	F	*p*
(1) Presence of sheath	1	14.0813	<0.001	(1) Presence of sheath	1	2.8779	>0.05
(2) Nematode species	1	13.8195	<0.001	(2) Part of glass tube	2	86.8856	<0.001
(3) Presence of nematodes	1	5.0018	<0.05	(3) Nematode species	1	1.9545	>0.05
(4) Spore type	1	7.0388	<0.01	(4) Time	2	1.1445	>0.05
1 × 2	1	9.5566	<0.01	1 × 2	2	6.0719	<0.01
1 × 3	1	4.8463	<0.05	1 × 3	1	0.4680	>0.05
2 × 3	1	6.8541	<0.01	2 × 3	2	3.6176	<0.05
1 × 4	1	4.5062	<0.05	1 × 4	2	0.0601	>0.05
2 × 4	1	9.3411	<0.01	2 × 4	4	15.9850	<0.001
3 × 4	1	2.6101	>0.05	3 × 4	2	0.2696	>0.05
1 × 2 × 3	1	4.9220	<0.05	1 × 2 × 3	2	9.6688	<0.001
1 × 2 × 4	1	4.3586	<0.05	1 × 2 × 4	4	2.4874	>0.05
1 × 3 × 4	1	2.4981	>0.05	1 × 3 × 4	2	0.0614	>0.05
2 × 3 × 4	1	4.7677	<0.05	2 × 3 × 4	4	2.8772	<0.05
1 × 2 × 3 × 4	1	2.9877	>0.05	1 × 2 × 3 × 4	4	3.7649	<0.01
error	84	2.8677	>0.05	error	36

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
