# Peer review of "Dissemination of Isaria fumosorosea Spores by Steinernema feltiae and Heterorhabditis bacteriophora"

_jof, 2020, doi:10.3390/jof6040359_

Round 1
Reviewer 1 Report
The paper is original and interesting, and the data properly analized. This work deserves to be published. One note: all the scientific names (fungi, nematodes, insects or mites), when reported for the first time in the text, should be written in full with Autority and systematics
Author Response
We appriciate the reviewers comments. The authorities and systematics were added to all organisms scientific names at first use (see the revised manuscript).
Jiri Nermut
Reviewer 2 Report
This is quite an interesting paper concerning 2 types of entomopathogens used in biological control of insect populations. The results indicate dissemination of Isaria fumosorosea spores by entomopathogenic nematodes, which can be useful in integrated pest management. Generally, the experiments were well designed. The introduction and discussion of the manuscript should be more concise. The Authors should also consider rearrangement of some sentences/paragraphs to improve its logical sequence.
Detailed comments
L42: sensitive sense organs - this is an assumption. There is little/no knowledge about the sensitivity of the organs of entomopathogenic nematodes.
L48: native host - this is an overstatement. Although G. mellonella larvae are usually used as bait for entomopathogenic nematodes or to breed them – it is still a surrogate host.
L56-57: sentence “Epizootics are… is – in my opinion - not necessary.
L69: “attacking” – I’d rather use “infect”.
L62: use “host” instead of “pest”
L87: indicate the order and family of Curculio caryae
L103: “evaluate and quantify” – use evaluate or quantify; “efficacy” – taking into account the achieved results, I would rather use “possibility”
L110: indicate the systematic affiliation of Steinernema and Heterorhabditis in the introduction
L161: too many technical details.
L195: you were trying to explain the effect of nematode species on fungus dispersal, not “differences between the nematode species”
L231: if there were no statistically significant differences between the dispersal of blastospores and conidia – do not suggest that something was higher
L236: the whole paper: do not indicate the exact P value (use P>0.05 or P<0.05/P<0.01/P<0.001).
L238: indicate the P value (if there were statistically significant differences)
Fig. 3. Indicate statistically significant differences between the columns in the graphs
L263-265: indicate only the number of CFU, not both the number of CFU and the percentage of the dose (it is not necessary and not informative)
Fig. 5: should be removed. The presented results are described. The graph is unclear and not informative (no statistically significant differences indicated).
Table 1. Compare the number of spores in the different glass tube sections. Indicate statistically significant differences (if applicable).
Figure 6. The arrows indicate some objects on the IJs surface. Presumably – blastospores. But bacterial rods look similar. This cannot be resolved. Remove the figure.
Table 2. Describe the presented results.
L290: there is no relation between the nematode efficacy in host searching/killing and the fungus dissemination.
L331: “highly” is an overstatement
L364: H. bacteriophora desheated by “abrasive environment” – it is only possible, not a fact (without a closer look under a microscope)
L386-393: there is little/no connection with the main topic. Remove the paragraph.
L395-400: as above.
Author Response
Dear reviewer,
We really appreciate the comments and suggestions to our manuscript. All the suggested changes were accepted and incorporated to the revised manuscript. We hope that the text is now more concise.
Best regards
Jiri Nermut
Reviewer 3 Report
Very nice study. Provide new information and innovative information. Nice experimental degsin.
Recommand to publish.
Only one comment: Fig. 5 is not clear. I suggest to present the data in a table
Author Response
Dear reviewer,
We appreciate the comments and suggestions. Figure 5 was removed. Data are actually presented in table 1 in form of CFUs. Percentage proportions that were presented in the figure 5 are not shown as they express exactly the same information as CFUs only in other form. We would like to avoid duplicity.
Thank you for your time
Jiri Nermut